# Mycotoxins in Maize Silage from China in 2019

**DOI:** 10.3390/toxins14040241

**Published:** 2022-03-27

**Authors:** Dawei Zhang, Liansheng Zhao, Yakun Chen, Heyang Gao, Yu Hua, Xianjun Yuan, Hailin Yang

**Affiliations:** 1Key Laboratory of Industrial Biotechnology, Ministry of Education, School of Biotechnology, Jiangnan University, Wuxi 214101, China; dawei.zhang@dsm.com; 2State Key Laboratory of Animal Nutrition, Institute of Animal Science, Chinese Academy of Agricultural Sciences, Beijing 100193, China; aaronann@163.com (L.Z.); chenyakun2021@163.com (Y.C.); 3Romer Labs Analytical Service (Wuxi) Ltd., No.6-1 Chunyu Road, Xishan District, Wuxi 214101, China; labcn@romerlabs.com (H.G.); fisher.hua@romerlabs.com (Y.H.); 4Institute of Ensiling and Processing of Grass, Nanjing Agricultural University, Weigang 1, Nanjing 210095, China; yuanxianjun@njau.edu.cn

**Keywords:** masked mycotoxins, emerging mycotoxins, maize silage, co-occurrence, UPLC-MS/MS

## Abstract

Animal feed (including forage and silage) can be contaminated with mycotoxins. Here, 200 maize silage samples from around China were collected in 2019 and analyzed for regulated mycotoxins, masked mycotoxins (deoxynivalenol, 3-acetyldeoxynivalenol, 15-acetyldeoxynivalenol, and deoxynivalenol-3-glucoside), and emerging mycotoxins (beauvericin, enniatins, moniliformin, and alternariol). Deoxynivalenol and zearalenone were detected in 99.5% and 79.5% of the samples, respectively. Other regulated mycotoxins were detected in fewer samples. The highest deoxynivalenol and zearalenone concentrations were 3600 and 830 μg/kg, respectively. The most commonly detected masked mycotoxin was 15-acetyldeoxynivalenol, which was detected in 68.5% of the samples and had median and maximum concentrations of 61.3 and 410 μg/kg, respectively. The emerging mycotoxins beauvericin, alternariol, enniatin A, enniatin B1, and moniliformin were detected in 99.5%, 85%, 80.5%, 72.5%, and 44.5%, respectively, of the samples but at low concentrations (medians <25 μg/kg). The samples tended to contain multiple mycotoxins, e.g., the correlation coefficients for the relationships between the concentrations of beauvericin and deoxynivalenol, deoxynivalenol and zearalenone, and zearalenone and beauvericin were 1.0, 0.995, and 0.995, respectively. The results indicated that there needs to be more awareness of the presence of one or more masked and emerging mycotoxins in maize silage in China.

## 1. Introduction

Mycotoxins are low-molecular-weight secondary fungal metabolites produced mainly by species of *Alternaria*, *Aspergillus*, *Fusarium*, and *Penicillium* [1]. The most common mycotoxins in animal feed and food for humans are aflatoxins, ochratoxin A (OTA), trichothecenes (type A (HT-2 and T-2 toxins) and type B (deoxynivalenol (DON), zearalenone (ZEN), and fumonisins B1 and B2 (FB1 and FB2), respectively)), and emerging mycotoxins (fusaproliferin, moniliformin (MON), beauvericin (BEA), and enniatins (ENNs)) [2]. Animal feed (including forage and silage) can be contaminated with mycotoxins in the field, pre- and post-harvest, during storage (including the ensiling process), and during feeding [3]. Mycotoxins can decrease feed intake and feed utilization and can also suppress immunity and cause economic losses [4].

Animals may frequently be concurrently exposed to multiple mycotoxins [5]. Mycotoxins can be toxic to farm animals and can cause distress and decrease productivity. Some mycotoxins can be transferred to animal products for human consumption such as meat, eggs, and milk [6]. For example, aflatoxins can have feed to milk transmission rates of 1–6% [7]. Many mycotoxigenic fungi are not specific to hosts; however, they are cereal crops grown under warm, damp, and humid conditions [8]. Each year mycotoxin contamination of cereal, fruit, and oilseed crops causes billions of dollars in losses world-wide due to reduced crop yields, lost trade revenues (local and international), livestock illnesses, and adverse human health effects. Economic losses due to mycotoxin contamination are estimated as billions of US dollars annually worldwide [9].

Silage is the main component of the diets of domestic ruminants, contributing 50–70% of dry matter intake [10]. Maize silage is one of the most important feedstuffs for ruminants in China and Europe because of its high nutritive value, palatability, and high biomass yield [11,12,13]. Ensiling involves anaerobic fermentation by lactic acid bacteria, which produce lactic acid and decrease the pH of the silage. This preserves the forage [14].

Molds and yeasts are undesirable microorganisms when preparing silage because they cause losses of dry matter and nutrients and produce mycotoxins. Contamination with mold mainly occurs in the field or during the ensiling process. Mycotoxins have been detected in maize silage around the world. In a survey, 300 samples of silage from 150 farms in Ireland were analyzed, and ENNs and BEA were detected in all of the silage samples collected over the 2-year period [15]. In another survey, 120 samples of silage were collected from farms in Poland in 2015 and both regulated and emerging mycotoxins were detected [11]. In 2018, 251 samples from 16 trench silos and three silage bags were collected from dairy farms in Spain, and regulated and masked mycotoxins were detected [10]. Maize silage was collected from 18 farms in Shandong Province and five important mycotoxins (DON, ZEN, AFB1, OTA, and T-2) were detected using an ELISA method [16]. Few surveys of mycotoxins in silage samples from across China have been performed. Most surveys of mycotoxins in maize silage have been focused on regulated mycotoxins rather than modified and emerging mycotoxins, which are difficult to detect using ELISA methods.

The objective of the study was to identify and quantify mycotoxin contamination of maize silage from across China by ultrahigh-performance liquid chromatography tandem mass spectrometry.

A total of 200 silage samples were collected from across China in 2019. The concentrations of 53 mycotoxins in the samples were determined using a multi-mycotoxin liquid chromatography-tandem mass spectrometry method.

## 2. Results

### 2.1. Mycotoxins in Maize Silage

Maize silage samples (*n* = 200) were collected from dairy cattle farms across China in 2019. Mycotoxin concentrations in the samples were determined by liquid chromatography tandem mass spectrometry. The detection rates, median concentrations, and maximum concentrations of the mycotoxins in the samples are shown in Table 1.

The concentrations of 53 mycotoxins of six types in the 200 maize silage samples were determined. At least three mycotoxins were found in each sample. The regulated mycotoxins aflatoxin B1, aflatoxin B2, aflatoxin G1, aflatoxin G2, and ochratoxin A were not detected in any of the samples. Ergot alkaloids were detected in only one sample, which was from Yunnan Province and contained ergot alkaloids at a concentration of 15.3 μg/kg. It was concluded that ergot alkaloids are rarely present in maize silage in China. The *Fusarium* toxins DON and ZEN were detected in 99.5% and 79.5% of the samples, respectively. The median and maximum DON concentrations were 315 and 3185 μg/kg, respectively. The fumonisins FB1, FB2, and FB3 were detected in 72%, 60%, and 32.5%, respectively, of the samples. The median and maximum concentrations were 61.6 and 558 μg/kg, respectively, for FB1, 31.4 and 198 μg/kg, respectively, for FB2, and 18.8 and 79.5 μg/kg, respectively, for FB3. Patulin was detected in one sample, and the concentration was relatively high (257 μg/kg). The maximum allowed patulin concentration in fruit and beverages in China is 50 μg/kg. However, patulin concentrations in animal feed are not regulated in China. Patulin concentrations in animal feed and feed products are also not regulated in the EU or USA [17,18].

15-Acetyldeoxynivalenol(15-ACDON) was detected in 68.5% of the samples, which was higher than any other trichothecene except DON. The highest nivalenol (NIV) concentration was 1302 μg/kg. NIV was detected in 10 samples, all from south China. T-2 and HT-2 were detected in 1.5% and 4.0%, respectively, of the samples, and the concentrations were 6.3 μg/kg and 31.6 μg/kg. 3-Acetyldeoxynivalenol(3-ACDON) was not detected in any of the samples. The median and maximum 15-acetoxyscirpenol concentrations were 68.5 and 80.0 μg/kg, respectively. 

Modified mycotoxins, including deoxynivalenol-3-glucoside, α-zearalenol (α-ZEL), and β-zearalenol (β-ZEL), were also analyzed. The concentrations of these modified mycotoxins were not higher than the concentrations of their parent toxins. Deoxynivalenol-3-glucoside was not detected in any of the samples, and α-ZEL and β-ZEL were detected in two samples. The median and maximum concentrations were 9.8 and 12.5 μg/kg, respectively, for α-ZEL and 15.7 and 23.6 μg/kg, respectively, for β-ZEL. The six most frequently detected mycotoxins were emerging mycotoxins. BEA was detected in 99.5% of the samples. Alternariol (AOH) was detected in 85% of the samples. Enniatin A (ENNA) was detected in 80.5% of the samples. Enniatin B1 (ENNB1) was detected in 72.5% of the samples. MON was detected in 44.5% of the samples. Enniatin B (ENNB) was detected in 35% of the samples. The concentrations of BEA and ENNA (315 and 307 μg/kg, respectively) were higher than the concentrations of the other emerging mycotoxins. Only four emerging mycotoxins (ENNA1, mycophenolic acid (MPA), sterigmatocystin (STG), and roquefortine C) were detected in <10% of the samples. 

Mycotoxins were found to co-occur in maize silage and showed in Figure 1. At least three mycotoxins were detected in each sample, and 13 mycotoxins were detected in one sample. The mean number of mycotoxins detected in each sample was 8.6, and 86.5% of the samples contained between seven and 11 mycotoxins. The most common co-occurring mycotoxins were DON and BEA, for which the correlation coefficient was 1.0. The next most common co-occurring mycotoxins were DON and ZEN and ZEN and BEA, for which the correlation coefficients were both 0.9950, and the next most common co-occurring mycotoxins were FB1 and DON and FB1 and BEA, for which the correlation coefficients were both 0.9497. 

### 2.2. Sources of the Samples

The samples were collected from the 18 main areas for dairy production in China. A total of 200 samples were collected from dairy farms in these areas. The provinces from which the samples were collected were divided into four regions, Central China (CT area) consist of Anhui (4), Shandong (22), Henan (24), and Hebei (24) provinces. North East (NE area) consisted of Heilongjiang (12), Jilin (7), Liaoning (8), and Inner Mongolia (6). South West (SW area) consisted of Guangxi (4), Guizhou (8), and Yunnan (8). North West (NW area) consisted of Gansu (14), Ningxia (18), Qinghai (7), Shanxi (16), Shanxi (6), and Xinjiang (12). Detailed information is shown in Figure 2. 

### 2.3. Regulated Mycotoxins in Maize Silage

Of the regulated mycotoxins, DON, ZEN, and fumonisins were found in the most samples (Figure 3). DON was detected in 100% of the samples from Central China (CT), Southwest China (SW), and Northwest China (NW) and in 97% of the samples from northeast China (NE). ZEN was detected in 95–100% of the samples from CT, SW, and NW and 84.8% of the samples from NE. DON and ZEN were co-occurring mycotoxins, possibly because of the climates and latitudes of the areas the samples were collected from. Fumonisins were also detected in a large proportion of the samples. FB1, FB2, and FB3 were detected in 87.8%, 82.4%, and 60.8%, respectively, of the samples. The highest DON concentration (3587 μg/kg) was found in a sample from NW. The highest ZEN concentration (831 μg/kg) was found in a sample from SW. The highest total fumonisins (FB1+FB2+FB3) concentration (812 μg/kg) was found in a sample from CT. Patulin was detected in only one sample, and ergocryptine was also detected in only one sample. 

### 2.4. Trichothecenes in Maize Silage

The concentrations of the trichothecenes T-2, HT-2, 15-acetoxyscirpenol (15-ACDAS), 15-ACDON, and NIV were determined. 15-ACDON was found in the most samples (between 65.0% and 72.7% of the samples from the different regions). Acetyl-DON was found in the same proportion of samples as the parent toxin. NIV was detected in samples from SW and NE. NIV was detected in 45.0% of the samples from SW, and the median and maximum concentrations were 453 and 1302 μg/kg, respectively. This indicated that NIV contamination is common in SW. NIV was detected in only one sample from NE, and the NIV concentration in that sample was 120 μg/kg. 15-ACDAS, T-2, and HT-2 were detected in few samples from NE and NW, and the concentrations were low. 

### 2.5. Masked Mycotoxins and Metabolites in Maize Silage

Mycotoxins such as deoxynivalenol can become conjugated with glucose moieties to form masked mycotoxins such as deoxynivalenol-3-glucoside (D3G), and ZEN can be transformed into α-ZEL and β-ZEL in cultures of suspended maize cells. These masked mycotoxins and metabolites are not readily detected using conventional methods. However, masked mycotoxins and metabolites can be deconjugated by enzymes in the digestive tracts of animals to release the parent mycotoxins. This study was focused on D3G, α-ZEL, and β-ZEL. D3G has been found to co-occur with DON in cereals. However, D3G was not detected in any of the samples analyzed in this study. Low concentrations of α-ZEL were detected in samples from SW and NW, and β-ZEL was only detected in samples from CT and SW. 

### 2.6. Emerging Mycotoxins and Other Mycotoxins

For all four regions, BEA was detected in 100% of the samples. ENNA was detected in 90.4% of the samples from NW. ENNA1 was detected in 9.6% of the samples from NW. ENNB and ENNB1 were detected in 54.8% and 95.9%, respectively, of the samples from NW.MON was detected in samples from all of the regions. MON was detected in 100% of the samples from CT but only 27.3% of the samples from NE. AOH was detected at high concentrations in the samples from all of the regions. AOH was detected in 91.9% of the samples from CT but in fewer of the samples from NW. MPA, STG, and roquefortine C were found in few samples. 

The BEA concentrations are shown in Table 2 The mean, median, and maximum BEA concentrations in the samples from NW were 28.2, 19.7, and 225 μg/kg, respectively. The lowest mean, median, and maximum BEA concentrations (18.5, 11.6, and 91.1 μg/kg, respectively) were for the samples from SW. 

The cyclic depsipeptides ENNs are produced by a wide range of *Fusarium* fungi, including *Fusarium acuminatum*, *Fusarium avenaceum*, *Fusarium oxysporum*, *Fusarium poae*, *Fusarium sambucinum*, *Fusarium sporotrichioides*, and *Fusarium tricinctum*. The most frequently detected ENNs were ENNA, ENNA1, ENNB, and ENNB1. As shown in Table 3, the highest ENNs concentrations were found in the samples from NW, and ENNA1 was not detected in the samples from SW.

The MON concentrations are shown in Table 4. MON was detected at relatively low concentrations. The highest MON concentration, 116 μg/kg, was found in a sample from NW. The highest mean and median MON concentrations, 21.0 and 10.9 μg/kg, respectively, were found in silage from SW. 

The STG and MPA concentrations are shown in Table 5. STG was detected in only one sample, at a concentration of 146 μg/kg. The highest median and maximum MPA concentrations, 95.6 and 143 μg/kg, respectively, were found for the samples from CT. 

The AOH and ZAN concentrations are shown in Table 6. The di-benzopyrone derivative AOH, which is produced by many *Alternaria* species, is cytotoxic and induces apoptotic cell death by affecting the mitochondria. AOH was detected in samples from all four regions. The highest AOH concentrations were found in samples from NW, and the mean, median, and maximum concentrations were 28.2, 19.7, and 225.0 μg/kg, respectively.

The nonsteroidal estrogenic mycotoxin ZEN is produced by *Fusarium* species, which colonize several grains. Zearalanol (ZAN) is a metabolite of ZEN. ZAN was detected in all of the samples in which ZEN was detected, but neither was detected in many of the maize silage samples. ZAN was detected in only four samples from NW, and the mean, median, and maximum concentrations were 7.3, 7.4, and 9.6 μg/kg, respectively. 

## 3. Discussion

*Fusarium* mycotoxins such as DON, ZEN, and fumonisins were found to be the most frequently occurring agriculture-related mycotoxins, which was also the case in previous studies. A total of 86 maize silage samples from China were analyzed in 2016, and the most common mycotoxins that were detected were DON and ZEN. Aflatoxin B1 was also detected in that study, but aflatoxins B1, B2, G1, and G2 were not detected in our samples. Aflatoxin B1 has previously been found to be common in China [19,20]. This is because maize silage samples were analyzed in most previous studies using ELISA methods, which can give false positive results, particularly for complicated matrices. Similar results have been reported in surveys of silage from Europe, Panasiuk et al. analyzed 120 silage samples from Poland and aflatoxins were not detected [11]. In the studies of Dr. Garon that the mature silage was normally could not detected aflatoxins was because that the ensiling process would break down the aflatoxins [21,22]. The *Fusarium* toxins DON and ZEN were two of the most frequently detected mycotoxins in the maize silage samples, being detected in 82% and 57%, respectively, of the samples. The mean DON and ZEN concentrations were 447 and 82.4 μg/kg, respectively [11]. A total of 158 maize silage samples from European dairy farms were analyzed in a previous study [23]. Aflatoxin B1 was not detected in any of the samples, but both DON and ZEN were present in 67.7% of the samples, they were the most prevalent mycotoxins in silages in the study.

Trichothecenes are mostly produced by species of the *Fusarium* genus, and some species in the *Myrothecium*, *Stachybotrys*, and *Trichoderma* genera also produce some Trichothecenes. Trichothecenes are found in cereal grains such as barley, maize, oats, rice, and wheat. Trichothecenes are sesquiterpenoids that contain characteristic epoxide groups [24]. Cereal products infected with DON-producing fungi readily become contaminated with NIV, acetylated derivatives (e.g., 3-acetyldeoxynivalenol (3-ACDON) and 15-ACDON), and the glucose moiety D3G. Mycotoxin derivatives, masked mycotoxins, cannot be detected using conventional analytical techniques because their structures are different from the parent mycotoxin structures. We found that 15-ACDON and NIV were the most common masked mycotoxins in maize silage in China. 15-ACDON and NIV were detected in 68.5% and 5%, respectively, of the samples. However, 3-ACDON and D3G were not detected. Masked mycotoxins have only been detected in wheat and corn samples from China, but no survey of masked mycotoxins in maize silage from China has yet been performed [25,26]. Masked mycotoxins have been detected in silage samples from other countries. For example, NIV, 3-ACDON, 15-ACDON, and D3G were detected in 59.5%, 0%, 5.1%, and 25.3% of silage samples from Europe in a previous study [23]. NIV was the most common masked mycotoxin in maize silage from Poland, and 3-ACDON was detected at concentrations of 24.9–37.2 μg/kg [11]. Similar results were found for samples from Spain [10]. 15-ACDON was detected in 10% of the samples, and the 15-ACDON concentrations were 2.44–6.58 μg/kg. Masked mycotoxins are of increasing interest in the food and feed safety field, and are always considered to be as toxic as the parent mycotoxins. During digestion in the gastrointestinal tract, enzymes will remove the glucose or acetyl moiety from a masked mycotoxin to release the parent mycotoxin. The sum of the trichothecenes concentrations has been used to assess mycotoxicity. There are no regulatory limits for NIV, 3-ACDON, and 15-ACDON in China. These results indicate that more attention should be paid to NIV, 3-ACDON, and 15-ACDON contamination of feed products to ensure that animal feed and animal products are safe. 

Emerging mycotoxins are a new group of mycotoxins including BEA, ENNs, AOH, and MON, all produced by the most common grain-contaminating fungi, *Fusarium* spp. Lately, much attention has been given to this new group of mycotoxins [27]. Emerging mycotoxins have been defined as “mycotoxins, which are neither routinely determined, nor legislatively regulated; however, the evidence of their incidence is rapidly increasing”. More emerging mycotoxins have been detected in agricultural products using modern analytical techniques. The dominant emerging mycotoxins in the maize silage samples were the *Fusarium* metabolites BEA, ENNs, and MON, the *Aspergillus* metabolite STG, the *Penicillium* metabolites MPA and roquefortine C, and the *Alternaria* metabolite AOH. The toxic precursor of the aflatoxin STG is closely structurally related to Aflatoxin B1. The International Agency for Research on Cancer has classed STG as a group 2B carcinogen (possibly carcinogenic to humans) [28]. MPA can suppress the immune system and has been associated with several sometimes-life-threatening viral infections. These emerging mycotoxins are not yet regularly analyzed and are not yet regulated. The basic principle of toxicology (the dose makes the poison) means that both the toxicity and proportion of samples these mycotoxins occur in are important when performing a risk assessment. BEA impairs the development of cultured porcine oocytes and early embryos. Exposure to BEA decreases progesterone synthesis in cumulus cells, decreases MDR1 activity by depleting ATP in zygotes, and decreases mitochondrial activity in early embryos [29]. BEA inhibits estradiol and progesterone synthesis in bovine granulosa cells by suppressing CYP19A1 and CYP11A1 gene expression [30]. The concentrations of BEA and ENNs in maize silage samples were similar to concentrations found in previous studies. It has previously been found that BEA is more commonly detected than other emerging mycotoxins. BEA was detected in 87% of samples (108 of 120 samples) from Poland, and the mean and maximum concentrations were 35.8 and 1309 μg/kg, respectively [11]. ENNB, ENNB1, ENNA1, and ENNA were detected in 89%, 78%, 71%, and 66% of the samples, respectively [11]. Similar results were found in a study of maize silage from Israel [13]. ENNA1 was the most commonly found enniatin mycotoxin, being found in 80% of the samples [13]. However, all of the enniatins were found at very low concentrations and BEA was detected in more samples than the enniatins [13]. No data are available to indicate whether BEA could be toxic to cattle at the concentrations detected in maize silage from China (15–30 μg/kg), but it has been found that BEA and Fumonisin B1 can jointly negatively affect reproduction in cattle [31]. AOH is produced by many *Alternaria* species. AOH was detected in 31% of feed and agricultural commodity samples from Europe, and the maximum concentration was 1840 μg/kg. AOH has been detected in 28.5% of samples from European countries [32]. AOH was detected in more of the samples we collected in China and at higher concentrations than in samples from the European countries This could have been caused by the different latitudes of European countries and China. European countries are in the northern temperate zone but most Chinese regions are in the subtropical zone. MON has regularly been detected in cereals from various parts of the world. It has been found that MON is very toxic in vivo. MON mainly affects the heart, causing acute heart failure, but can also cause muscle weakness, respiratory distress, and weakened immunity and performance [20]. We detected MON in 44.5% of the samples from China. This was similar to the results of a previous study in which MON was detected in 44.9% of samples from European countries [23]. MON was also detected in 46.7% of 30 maize and wheat silage samples from Israel [13].

## 4. Conclusions

Masked mycotoxins, particularly 15-ACDON, and the emerging mycotoxins BEA, ENNs, MON, and AOH were detected in many maize silage samples from China. The *Fusarium* toxins DON, ZEN, and FB1 were also frequently detected. Mycotoxin occurrence data for different areas of China will allow regional mycotoxin patterns to be understood and provide data for improving methods for controlling and preventing mycotoxin production in maize silage. Mycotoxin contamination may occur pre- or post-harvest. For maize silage, post-harvest includes the ensiling process. Further research should be performed to investigate changes in mycotoxin concentrations pre- and post-harvest and particularly during fermentation.

## 5. Materials and Methods

### 5.1. Sample Collection

A total of 200 samples were collected from farms in China in 2019. The samples were from 100 farms in 17 provinces.

### 5.2. Silage Sampling

A nine-point sampling pattern was used. Samples were collected from 30–40 cm below the silage surface, 30–40 cm further down, and 30–40 cm above the bottom. Samples were collected from points separated horizontally by 30–50 cm. At each sampling point, a 300–500-g sample was collected. The samples from all nine points were then homogenized in a bin. A 500-g subsample was then removed for mycotoxin analysis.

The silage sampling was carried out from May to June in 2019 (9 to 10 months after ensiling), about 200 farms were surveyed. They consisted of 74,33,20 and 73 farms from Central China (CT area), North East (NE area), South West (SW area), and North West (NW area) of China, respectively. The 200 silages sampled consisted of banker (*n* = 185), pile (*n* = 4) and round-baled silages (*n* = 11). The banker and pile silos were packed by tractor to achieve about 240 kg/m^3^. The DM contents ranged between 20.46~41.35% with a average DM content of 30.10%.5.3. Analytical Method.

#### 5.2.1. Sample Extraction and Preparation

Ultrahigh-performance liquid chromatography tandem mass spectrometry analysis was performed using an Agilent 1290 II liquid chromatograph (Agilent Technologies, Santa Clara, CA, USA) coupled to a Sciex Qtrap 5500 mass spectrometer (SCIEX, Framingham, MA, USA) with an electrospray ionization source. The method was developed by Romer Labs Austria and validated by Romer Labs Analytical Service (Wuxi) Ltd. The method was introduced to the market by Romer Labs as the Spectrum Top 50 in 2018. (We have applied for the patent for invention). Briefly, 10 g of a ground sample was added to a 50-mL centrifuge tube and extracted with 30 mL of a 79:20:1 *v*/*v*/*v* mixture of acetonitrile (Merck, Frankfurt, Germany), water (Merck, Frankfurt, Germany), and formic acid (Sigma–Aldrich, City of Saint Louis, MO, USA) for 60 min. The sample was blended for 60 min using aRO 500 system (C.Gerhardt, Königswinter, Germany). The sample was then centrifuged for 4 min at 2000× *g* using a 3–15 centrifuge (Sigma, Yangzhou, China). A 100-µL aliquot of the supernatant was transferred to a glass vial and then diluted with 600 μL of the diluent (10% eluent A (97:2:1 *v*/*v*/*v* MeOH (Merck, Frankfurt, Germany), /H_2_O (Merck, Frankfurt, Germany), /acetic acid (Merck, Frankfurt, Germany), 5 mM of ammonium acetate (Sigma–Aldrich, City of Saint Louis, MO, USA) and 90% eluent B (10:89:1 *v*/*v*/*v* MeOH/H_2_O/acetic acid, 5 mM of ammonium acetate)). The vial was vortexed for 5 s and then analyzed by liquid chromatography tandem mass spectrometry. Separation was achieved using a binary gradient elution profile using eluents A and B. The flow rate was 0.5 mL/min. The column was a Gemini C18 column (100-mm long, 3.0-mm i.d., 3-µm particle size) with a C18 guard cartridge (4-mm long, 3-mm i.d.) (Phenomenex, Torrance, CA, USA). The gradient elution profile is shown in Table 7.

The injection volume was 9 µL of a sample and 1 µL of a mixture of mycotoxin internal standards. Qualification and quantification of each mycotoxin were performed in scheduled MRM mode using the Qtrap 5500 liquid chromatograph tandem mass spectrometer. External calibration was achieved using multi-analyte working solutions prepared by mixing different mycotoxin standards and diluting the mixture with a solvent. All the standard and internal standard was used was provided by (Romer Labs, Tulln, Austria). The typical samples with contamination of mycotoxins were shown in Figure 4.

#### 5.2.2. Method Validation

The parameters taken into account for method validation were linearity, sensitivity including limit of detection and limit of quantification, accuracy (recovery), and precision (repeatability and reproducibility), and ring test with four different labs using the same method around the world.

Sensitivity was performed by limit of detection (LOD) and limit of quantification (LOQ) values. The LOD and LOQ were calculated based on signal-to-noise ratio (S/N) of 3:1 and 10:1. The linearity were carried out with internal standard quantification (23 compound with internal standard) and external standard quantification (30 compound without internal standard).

Recoveries and relative standard deviation (RSDs) of mycotoxins were measured to validate the dilute and shoot method by spiking the blank samples at two different concentration (low level: 2LOQ; and high level 10LOQ) and then measuring them in 3 triplicates per day in 3 days. The precision of the method was determined by the repeatability (*n* = 3) and reproducibility (*n* = 9) studies and expressed the recovery (60.08~128.83%) and the relative standard deviation (RSD range 0.85~12.63%).

### 5.3. Data Analysis

The software MultiQuant 3.0.2 provided along with LCMSMS was used for peak review and calibration. Statistical analyses were performed using EXCEL 2017 and SPSS 21.0 software. The mean number of samples in which each mycotoxin was detected, and the mean, relative standard deviation, standard deviation, and quartile concentrations were calculated.

## Figures and Tables

**Figure 1 toxins-14-00241-f001:**
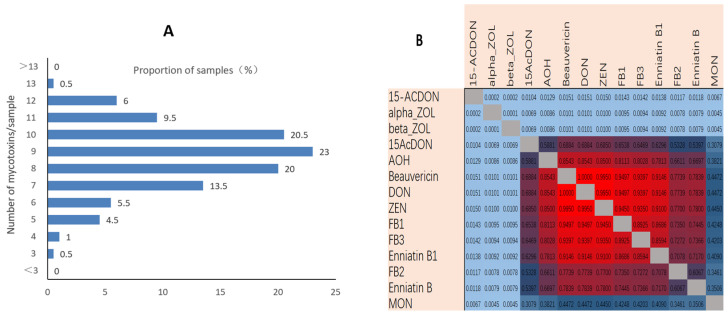
(**A**) Number of mycotoxins detected per sample. (**B**) The most relevant co-occurring mycotoxins.

**Figure 2 toxins-14-00241-f002:**
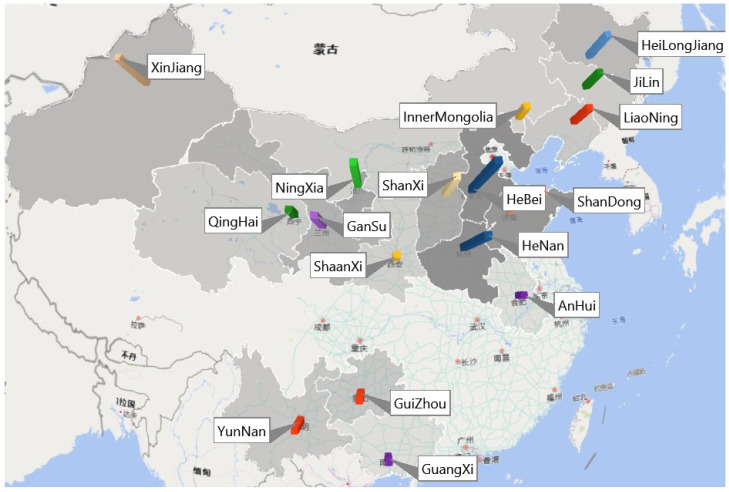
Map of China indicating where the maize silage samples were collected. (bar graph shows the amount of samples in each province. e.g., 蒙古: Mongolia; 不丹: Bhutan; 缅甸: Myanmar; 北京: Beijing; 渤海: Bohai; 呼和浩特: Hohhot; 武汉: Wuhan; 南昌: Nanchang; 成都: Chengdu and so on).

**Figure 3 toxins-14-00241-f003:**
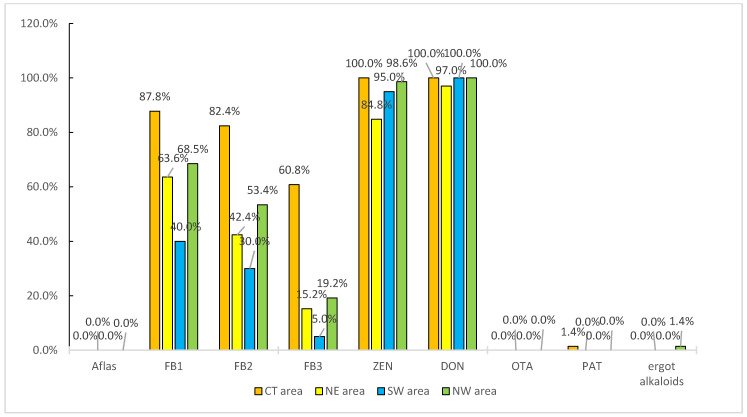
Occurrences of regulated mycotoxins in the samples from each area.

**Figure 4 toxins-14-00241-f004:**
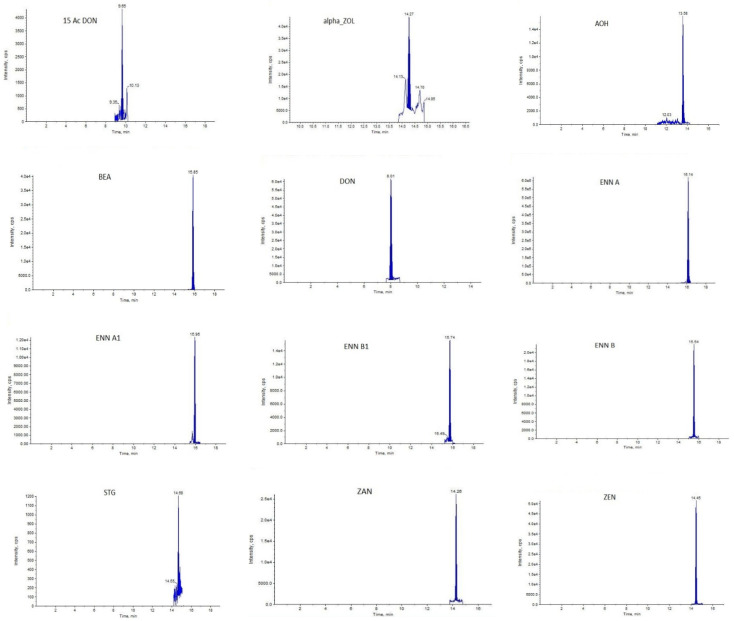
Typical mycotoxins chromatograph detected in maize silage samples.

**Table 1 toxins-14-00241-t001:** Detection rates and concentrations of mycotoxins in 200 silage samples collected in China in 2019. The mycotoxin concentrations are in μg/kg fresh silage. The numbers in bold are the largest five values in the relevant category (e.g., the highest numbers of positive samples or the highest medians).

Mycotoxin	Positive Samples (*n*)	Positive Samples (%)	Median Concentration (μg/kg)	75th Percentile (μg/kg)	95th Percentile (μg/kg)	Maximum Concentration (μg/kg)
*Regulated mycotoxins (except ergot alkaloids)*						
Aflatoxin B1 (AFB1)	0	0	-	-	-	-
Aflatoxin B2 (AFB2)	0	0	-	-	-	-
Aflatoxin G1 (AFG1)	0	0	-	-	-	-
Aflatoxin G2 (AFG2)	0	0	-	-	-	-
Deoxynivalenol (DON)	199	99.5	315	611	1884	3587
Fumonisin B1 (FB1)	144	72.0	61.6	121	261	558
Fumonisin B2 (FB2)	120	60.0	31.4	54.6	108	198
Fumonisin B3 (FB3)	65	32.5	18.8	30.3	49.7	79.5
Ochratoxin A (OTA)	0	0	-	-	-	-
Zearalenone (ZEN)	159	79.5	38.7	82.7	201	832
Patulin (PAT)	1	0.5	257	257	257	257
*ergot alkaloids*						
Agroclavine	0	0	-	-	-	-
Ergine	0	0	-	-	-	-
Ergocornine	0	0	-	-	-	-
Ergocorninine	0	0	-	-	-	-
Ergocristine	0	0	-	-	-	-
Ergocristinine	0	0	-	-	-	-
Ergocryptine	1	0.5	15.3	15.3	15.3	15.3
Ergocryptinine	0	0	-	-	-	-
Ergometrine	0	0	-	-	-	-
Ergometrinine	0	0	-	-	-	-
Ergosine	0	0	-	-	-	-
Ergotamine	0	0	-	-	-	-
Dihydrolysergol	0	0	-	-	-	-
Elymoclavine	0	0	-	-	-	-
*Type-A trichothecenes*						
Diacetoxyscirpenol (DAS)	0	0	-	-	-	-
HT-2 Toxin (HT-2)	8	4.0	31.6	49.4	64.2	68.5
Neosolaniol (NEO)	0	0	-	-	-	-
T-2 Toxin (T-2)	3	1.5	6.3	6.3	6.3	6.3
*Type-B trichothecenes*						
15-Acetoxyscirpenol (15-ACDAS)	3	1.5	68.5	74.3	79.0	80.2
15-Acetyldeoxynivalenol (15-ACDON)	137	68.5	61.3	89.9	247	411
3-Acetyldeoxynivalenol (3-ACDON)	0	0	-	-	-	-
Nivalenol (NIV)	10	5.0	362	996	1253	1302
Fusarenon X (FUX)	0	0	-	-	-	-
*Modified mycotoxins*						
Deoxynivalenol-3-Glucoside (D3G)	0	0	-	-	-	-
α-Zearalenol (α-ZEL)	2	1.0	9.8	11.1	12.2	12.5
β-Zearalenol (β-ZEL)	2	1.0	15.7	20.1	23.6	24.5
*Emerging mycotoxins*						
Alternariol (AOH)	170	85.0	17.0	29.8	87.4	225
Beauvericin (BEA)	199	99.5	22.3	37.3	130	315
Enniatin A (ENNA)	161	80.5	20.6	30.1	49.1	307
Enniatin A1 (ENNA1)	11	5.5	1.56	2.22	4.34	5.08
Enniatin B (ENNB)	70	35.0	2.09	8.54	37.1	84.1
Enniatin B1 (ENNB1)	145	72.5	1.98	3.15	14.7	56.0
Moniliformin (MON)	89	44.5	5.21	11.7	55.0	116
Mycophenolic acid (MPA)	3	1.5	48.3	95.6	133	143
Sterigmatocystin (STG)	1	0.5	146	146	146	146
Penicillin Acid (PEA)	0	0	-	-	-	-
Roquefortine C (ROC)	3	1.5	10.3	14.2	17.3	18.0
T-2 Triol	0	0	-	-	-	-
T-2-Tetraol	0	0	-	-	-	-
Gliotoxin	0	0	-	-	-	-
Ochratoxin B (OTB)	0	0	-	-	-	-
Zearalanol (ZAN)	4	2.0	7.43	8.28	9.32	9.58

**Table 2 toxins-14-00241-t002:** Beauvericin (BEA) concentrations (μg/kg) detected in the maize silage samples. (Average and median values were calculated only in positive samples).

Regions	BEA(μg/kg)		
	Average	Median	Maximum
Central China (CT area)	25.4	14.0	152.0
Northeast (NE area)	24.2	16.2	128.0
Southwest (SW area)	18.5	11.6	91.1
Northwest (NW area)	28.2	19.7	225.0

**Table 3 toxins-14-00241-t003:** Enniatins (ENNA, ENNA1, ENNB, and ENNB1) concentrations (μg/kg) detected in the maize silage samples. (Average and median values were calculated only in positive samples).

Regions	ENNA			ENNA1			ENNB			ENNB1		
	Average	Median	Maximum	Average	Median	Maximum	Average	Median	Maximum	Average	Median	Maximum
CT area	18.9	15.0	55.4	1.23	1.23	1.23	2.01	1.55	3.44	1.68	1.36	6.00
NE area	24.2	23.1	55.0	2.63	1.50	5.08	2.74	1.52	17.3	3.34	2.19	27.8
SW area	27.0	18.4	91.1	ND	ND	ND	15.5	2.86	84.1	4.23	2.04	31.5
NW area	29.5	22.3	307	2.03	1.56	3.60	9.54	3.03	64.8	5.99	2.50	56.0

**Table 4 toxins-14-00241-t004:** Moniliformin (MON) concentrations (μg/kg) detected in the maize silage samples. (Average and median values were calculated only in positive samples).

Regions	MON		
	Average	Median	Maximum
CT area	12.8	5.6	98.7
NE area	10.5	5.2	35.9
SW area	21.0	10.9	99.1
NW area	10.2	3.8	116.0

**Table 5 toxins-14-00241-t005:** Sterigmatocystin (STG) and mycophenolic acid (MPA) concentrations (μg/kg) detected in the maize silage samples. (Average and median values were calculated only in positive samples, * ND means not detected).

Regions	STG			MPA		
	Average	Median	Maximum	Average	Median	Maximum
CT area	ND *	ND *	ND *	95.6	95.6	143
NE area	ND *	ND *	ND *	41.5	41.5	41.5
SW area	146.0	146.0	146.0	ND *	ND *	ND *
NW area	ND *	ND *	ND *	ND *	ND *	ND *

**Table 6 toxins-14-00241-t006:** Alternariol (AOH) and zearalanol (ZAN) concentrations (μg/kg) detected in the maize silage samples. (Average and median values were calculated only in positive samples, * ND means not detected).

Regions	AOH			ZAN		
	Average	Median	Maximum	Average	Median	Maximum
CT area	25.4	14.0	152.0	ND *	ND *	ND *
NE area	24.2	16.2	128.0	ND *	ND *	ND *
SW area	18.5	11.6	91.1	ND *	ND *	ND *
NW area	28.2	19.7	225.0	7.3	7.4	9.6

**Table 7 toxins-14-00241-t007:** Liquid chromatography conditions.

**Column**	Phenomenex Gemini C18 (3.0 × 100 mm; 3 µm) with suitable pre-column (optional)
Gradient	0 min–20% B 2 min–20% B 8 min–65% B 10 min–80% B 11 min–95% B 13 min–95% B 13.1 min–20% B 17 min–20% B
Temperature	45 °C
Flow rate	0.5 mL/min
Injection volume	Sample: 10 µL 13C-iStd-Mix: 2 µL
Injection program	Wash needle in Flushport for 15 s Draw 2 µL from Vial 1 Wash needle in Flushport for 15 s Draw def. amount of sample Wash needle in Flushport for 15 s Inject
Running Time	17 min

## Data Availability

Not applicable.

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
