# Peer review of "Mycotoxins in Maize Silage from China in 2019"

_toxins, 2022, doi:10.3390/toxins14040241_

Round 1
Reviewer 1 Report
The work is very interesting and has certainly required a lot of work. However, this reviewer thinks that a very important part of the work is missing, namely the description of the validation of the method and how the 53 different compounds were identified. The addition of one/two chromatograms would make the method clearer. The precise description of the method is indispensable in a paper where the focus is on the concentrations detected using that analytical method.
This reviewer is not a native English speaker, but we recommend that the entire manuscript be read by a native English speaker.
All requested changes are reported in the attached manuscript.

Author Response
Dear Editors and Reviewers:
Thank you for your letter and for the reviewers’ comments concerning our manuscript entitled “Masked and Emerging Mycotoxins in Maize Silage from China in 2019 (Toxins-1598133)” . Those comments are all valuable and very helpful in revising and improving our paper, as well as important guiding significance to our further researches. We have read thoroughly all comments and have made corrections with red color in the revision. The main list of corrections in the paper and the responses to the reviewer’s comments as following:
Editor
Reviewer #1
1. Regulated, Masked
A:We have revised the title as the reviewer’s suggestion.
2.Reference numbers of in-text citations should be before the full stop
A: We have put reference numbers of in-text citations before the full stop
3.L 39: the word "certain" appears four times should be rephrased
A:We have rephrased this sentence.
4.L 52: "in both year" should be rephrased
A: we have rephrased this sentence.
5.L 67: Insert the subsection "Assay validation" in which to report the results of the parameters evaluated (see also comment inserted on line 345). Also include a chromatogram
A:We had applied for the patent of invention of this method, so we do not want too many method details in this paper, but we showed the STD chromatograph and Samples chromatograph in the paper and demonstrate how we do the validation.
6.Table 1, In my opinion, it would be helpful to put the abbreviations used for each compound in this table, next to the compound name, this would make the results and discussion easier to read
A: We have added the abbreviations for the main compound in the table.
7.L 38: " 1%–6%" Please check this throughout the text
A: We have revised it and checked others throughout the text.
8.L 94: " 80.0μg/kg" Please check this throughout the text
A: We have added the space and checked this throughout the text
9.L 155-160: Standardize alpha and beta line 98(,) in Section 2.5
A: We have standardized the abbreviations of alpha and beta in line 98 and section 2.5
10. Fusarium, Fusarium acuminatum, Fusarium avenaceum, Fusarium oxysporum, Fusarium poae, Fusarium sambucinum, Fusarium sporotrichioides, and Fusarium tricinctum. Should be in italics
A: We have revised them as italic.
11. the uniforms of AVG, MED, MAX in table 4 should as in table 3,5,6,7
A: We have uniformed it for table 4.
12. L231: "67.7% and 67.7%, respectively" should be rephrased
A:We have revised this sentence.
13. L233- 235: This sentence should be rephrased
A:We have rephrased this sentence.
14. L258- 261: This sentence should be rephrased
A:We have repharsed this sentence.
15. L320: Insert the subsection "chemicals and reagents" in which to report all the solvents used (for extraction procedure, LC analysis..) and the reference standards, with the name of the companies that produced and supplied them
A:We have added in the text.
16. Table 8: in text the value below 9+1µL where 20µL in table
A:10uL is the correct one, we have revised.
17. L345: Insert a subsection "Quantification and performance evaluation" in which to report how the quantification of analytes and the evaluation of method performance(validation) have been carried out
A:Add 5.3.2 method validation
18. References must be reported in accordance with the journal style.
A: We have reported references in accordance with the journal style,
19.α-zearalenol, and β-zearalenol, should be α-ZEL and β-ZEL
A:We have changed the form of α-zearalenol, and β-zearalenol to α-ZEL and β-ZEL.

Reviewer 2 Report
The article presented, although it provides data of interest, cannot be accepted in its current form.
The main defect is not making reference to other previous articles carried out by various authors from different countries that have analyzed mycotoxins in corn. And in the discussion a critical comparison should be made.
On the other hand, various formal aspects are presented:
- "Masked and emerging" should be removed from the title
- The paragraph between lines 40 and 42 should be eliminated because it is very old, today that figure (25%) must be raised to close to 90%.
- In table 1, mycotoxins with a value of 0 or - must be eliminated. The text explains what mycotoxins are.
- Section 2.6 should not be subdivided.
- Table 2 and figure 7 are repetitive, one of the two must be eliminated.
- Figures 3, 4, 5, 6 and 8 must be removed. The explanations in the text are sufficient.
- The paragraph between lines 259 and 261 appears in the textbooks for high school graduates, it does not deserve to be in a scientific article and must be eliminated.
- Citation 9 should be deleted.
- In several bibliographic citations (12, 13, 19, 27) appears in the list of authors "etc." Authors must be included.
Author Response
Dear Editors and Reviewers:
Thank you for your letter and for the reviewers’ comments concerning our manuscript entitled “Masked and Emerging Mycotoxins in Maize Silage from China in 2019 (Toxins-1598133)” . Those comments are all valuable and very helpful in revising and improving our paper, as well as important guiding significance to our further researches. We have read thoroughly all comments and have made corrections with red color in the revision. The main list of corrections in the paper and the responses to the reviewer’s comments as following:
Reviewer #2
20. The article presented, although it provides data of interest, cannot be accepted in its current form. The main defect is not making reference to other previous articles carried out by various authors from different countries that have analyzed mycotoxins in corn. And in the discussion a critical comparison should be made.
A:The corn silage mycotoxin contamination is different from the mycotoxin coin corn, that’s why in the existing survey there are plenty of corn mycotoxin survey, but few maize silages survey. We have a lot of data in our lab and the mycotoxin distribution are different. If you insist these different matrices data compassion should be include in this review I will also add later.
21. "Masked and emerging" should be removed from the title
A: We have rephrased the title in the revision.
22. The paragraph between lines 40 and 42 should be eliminated because it is very old, today that figure (25%) must be raised to close to 90%.
A: We have rephrased the sentence in the revision.
23. In table 1, mycotoxins with a value of 0 or - must be eliminated. The text explains what mycotoxins are.
A: We have deleted the mycotoxins with a value of 0.
24. Section 2.6 should not be subdivided.
A:We have combined these subsections
25. Table 2 and figure 7 are repetitive, one of the two must be eliminated.
A:We have deleted Table 2 and add some of the sentence to explain the four regions.
26. Figures 3, 4, 5, 6 and 8 must be removed. The explanations in the text are sufficient.
A:We have deleted 3,4,5,6 but keep the Figure 8 in order to make it clear for the sampling pattern.
27. The paragraph between lines 259 and 261 appears in the textbooks for high school graduates, it does not deserve to be in a scientific article and must be eliminated.
A:We have rephrased this sentence.
28. Citation 9 should be deleted.
A:We replaced this citation with a new and related reference.
29. In several bibliographic citations (12, 13, 19, 27) appears in the list of authors "etc." Authors must be included.
A:We have unformed the references as the guidelines of the Journal.

Reviewer 3 Report
I appreciate the level of detail they include of their extraction & analytical methods. I’d also like to see some representative chromatograms of samples with multiple mycotoxins.
The methods of sample collection are disappointing. We are given information about where in China the samples come from and where in the silage stack / silage pit the samples come from, but we don’t know anything about the silage itself: What stage of growth /what % moisture is corn cut to make silage on these farms? Is this silage stored in covered pits / trenches or in open piles? What packing process is used to exclude excess air? How long after harvest was this stored before sampling for this study?
I’m surprised that the work from Audenaert & Haesaert or Garon are not included & discussed.
Specific points:
I see a problem with calculations on several of the tables. For example, Sterigmatocyctin was detected in only 1 sample, at 146 ug/kg. There is no way that the average was also 146 and the median was 146.
Line 211-215 and Table 7: This ZEN / ZAN point is unclear. “ZAN” is not mentioned in until this paragraph, and the full name is not given in the text – only in the title for Table 7.
Table 7. You measured mycotoxins in 73 samples in the NW area. You state that ZAN was present in only 4 of these samples. This means that you would have had 69 samples with zero ZAN, so the median would be zero.
Author Response
Dear Editors and Reviewers:
Thank you for your letter and for the reviewers’ comments concerning our manuscript entitled “Masked and Emerging Mycotoxins in Maize Silage from China in 2019 (Toxins-1598133)” . Those comments are all valuable and very helpful in revising and improving our paper, as well as important guiding significance to our further research. We have read thoroughly all comments and have made corrections with red color in the revision. The main list of corrections in the paper and the responses to the reviewer’s comments as following:
Reviewer #3
30. The methods of sample collection are disappointing. We are given information about where in China the samples come from and where in the silage stack / silage pit the samples come from, but we don’t know anything about the silage itself: What stage of growth /what % moisture is corn cut to make silage on these farms? Is this silage stored in covered pits / trenches or in open piles? What packing process is used to exclude excess air? How long after harvest was this stored before sampling for this study?
A:The silage sampling was carried out from May to June in 2019 (9 to 10 months after ensiling), about 200 farms were surveyed. They consisted of 74,33,20 and 73 farms from Central China (CT area), North East (NE area), South West (SW area) and North West (NW area) of China, respectively. The 200 silages sampled consisted of banker (n = 185), pile (n = 4) and round-baled silages (n = 11). The banker and pile silos were packed by tractor to achieve about 240 kg /m3. The DM contents ranged between 20.46~41.35% with a average DM content of 30.10%. We added in the review.
31. I’m surprised that the work from Audenaert & Haesaert or Garon are not included & discussed.
A:I have added two papers from Prof. Garon in the discussion and reference [26] and [27].
32. I see a problem with calculations on several of the tables. For example, Sterigmatocyctin was detected in only 1 sample, at 146 ug/kg. There is no way that the average was also 146 and the median was 146.
A:We are very sorry for the misunderstanding for you. All the mean and median values in this paper is positive mean value and positive median value. I have noted in all the chart.
33. Line 211-215 and Table 7: This ZEN / ZAN point is unclear. “ZAN” is not mentioned in until this paragraph, and the full name is not given in the text – only in the title for Table 7.
A: ZEN is zearalenone, and it was explained in the introduction section.ZAN is zearalanol. We have added the full name of ZAN in the text.
34. Table 7. You measured mycotoxins in 73 samples in the NW area. You state that ZAN was present in only 4 of these samples. This means that you would have had 69 samples with zero ZAN, so the median would be zero.
A:We are very sorry for the misunderstanding for you. All the mean and median values in this paper is positive mean value and positive median value. I have noted in all the chart.

Round 2
Reviewer 2 Report
The article has improved in its new presentation. However: - The title of the article should be "Mycotoxins in maize silage from China in 2019" - Figure 8 should be eliminated as it is a logical sampling (symmetrical) and perfectly explained in the text. - The new figures 6 and 7 do not make any substantial contribution and should be removed.
Author Response
Dear Editors and Reviewers:
Thank you for your letter and for the reviewers’ comments concerning our manuscript entitled “Mycotoxins in Maize Silage from China in 2019 (Toxins-1598133)” . Those comments are all valuable and very helpful in revising and improving our paper, as well as important guiding significance to our further research. We have read thoroughly all comments and have made corrections with red color in the revision. The main list of corrections in the paper and the responses to the reviewer’s comments as following:
1.The title of the article should be "Mycotoxins in maize silage from China in 2019"
A. We have revised the title as request.
2. Figure 8 should be eliminated as it is a logical sampling (symmetrical) and perfectly explained in the text.
A. We have deleted the figure 8.
3. The new figures 6 and 7 do not make any substantial contribution and should be removed.
A. We have deleted the figures 6 and 7.
